# RANDOM BIAS INITIALIZATION IMPROVING BINARY NEURAL NETWORK TRAINING

## ABSTRACT

Edge intelligence especially binary neural network (BNN) has attracted considerable attention of the artificial intelligence community recently. BNNs significantly reduce the computational cost, model size, and memory footprint. However, there is still a performance gap between the successful full-precision neural network with ReLU activation and BNNs. We argue that the accuracy drop of BNNs is due to their geometry. We analyze the behaviour of the full-precision neural network with ReLU activation and compare it with its binarized counterpart. This comparison suggests random bias initialization as a remedy to activation saturation in full-precision networks and leads us towards an improved BNN training. Our numerical experiments confirm our geometric intuition.

## 1 INTRODUCTION

Convolutional neural network has become one of the most powerful tools for solving computer vision, natural language processing, speech recognition, machine translation, and many other complex tasks. The most successful and widely-used recipe for deep neural network is ReLU-style activation function with MSRA style weight initialization (He et al., 2015). The standard sigmoid and the hyperbolic tangent were the most common activation functions, before the introductio of ReLU. ReLU-like activation functions are widely proved to be superior in terms of accuracy and convergence speed.

It is more common to use low-bit quantized networks such as Binary Neural Networks (BNNs) (Courbariaux et al., 2016) to implement such deep neural networks on edge devices such as cell phones, smart wearables, etc. BNNs only keeps the sign of weights and compute the sign of activations $\{-1, +1\}$ by applying Sign function in the forward pass. In backward propagation, BNN uses Straight-Through-Estimator (STE) to estimate the backward gradient through the Sign function and update on full-precision weights. The forward and backward loop of a BNN, therefore, becomes similar to the full-precision neural network with hard hyperbolic tangent *htanh* activation. The htanh function is a piece-wise linear version of the nonlinear hyper-bolic tangent, and is known to be inferior in terms of accuracy compared to ReLU-like activation function.

We examine a full-precision network with htanh activation to provide a new look in improving BNN performance. We conclude that the bias initialization is the key to mimic ReLU geometric behavior in networks with htanh activation. This conclusion challenges the common practice of deterministic bias initialization for neural networks.

Although the analysis is based on htanh function, this conclusion equally applies to BNNs that use STE, a htanh-like, back propagation scheme. Other saturating activations like hyperbolic tangent and sigmoid commonly applied in recurrent neural networks may benefit from this resolution as well.

Our novelties can be summarized in four items i) we analyze the geometric properties of ReLU and htanh activation. This provides an insight into the training efficiency of the unbounded asymmetric activation functions such as ReLU. ii) we propose bias initialization strategy as a remedy to the bounded activations such as htanh. iii) We back up our findings with experiments on full-precision to reduce the performance gap between htanh and ReLU activations. iv) We show this strategy also improves BNNs, whose geometric behavior is similar to the full-precision neural network with htanh activation.

There are very few works that focus on the initialization strategy of the bias term of the neural network. To the best of our knowledge, we are the first to propose random bias initialization as a remedy to the saturating full-precision neural network, also as a method to improve BNN training.

## 2 RELATED WORKS

(Glorot et al., 2011) proposed training deep neurals network with ReLU activation, and argued that ReLU activation alleviates the vanishing gradient problem and encourages sparsity in the model. The hyperbolic tangent only allowed training of shallow neural networks.

Since AlexNet (Krizhevsky et al., 2012), almost every successful neural network architectures use ReLU activation or its variants, such as adaptive ReLU, leaky ReLU, etc. Although many works reported that ReLU activation outperforms the traditional saturating activation functions, the reason for its superior performance remains an open question.

(Ramachandran et al., 2017) utilized automatic search techniques on searching different activation functions. Most top novel activation functions found by the searches have an asymmetric saturating regime, which is similar to ReLU. Farhadi et al. (2019) adapts ReLU and sigmoid while training. To improve the performance of saturating activations, Xu et al. (2016) proposed *penalized tanh* activation, which introduces asymmetric saturating regime to tanh by inserting leaky ReLU before tanh. The penalized tanh could achieve the same level of performance as ReLU activating CNN. It is worth to mention that similar ideas also appear in the related works of binarized neural network.

Gulcehre et al. (2016) improved the performance of saturating activations by adding random noise when the neuron is saturated, so the backward signal can easily pass through the whole model, and the model becomes easier to optimize. In this works, we proposed to randomize the non-saturated regime by using random bias initialization. This initialization can guarantee all backward signals can pass through the whole model equally.

The initial work on BNN appeared in Courbariaux et al. (2016), which limits both weights and activations to $-1$ and $+1$, so the weighted sum can be computed by bit-wise XNOR and PopCount instructions. This solution reduces memory usage and computational cost up to 32X compared with its full-precision counterpart. In the original paper, BNN was tested on VGG-7 architecture. Although it is an over-parameterized architecture for CIFAR 10 dataset, there is a performance gap between BNN and full-precision with ReLU activation. We believe the different between the two activations, BNN using the sign and full-precision using ReLU, is partially responsible for this gap.

XNOR-Net (Rastegari et al., 2016) developed the idea of BNN and proposed to approximate the full-precision neural network by using scaling factors. They suggest inserting non-Binary activation (like ReLU) after the binary convolution layer. This modification helps training considerably. Later, Tang et al. replaced replacing ReLU activation with PReLU in XNOR-Net to improve the accuracy. Note that XNOR-Net and many relaated works require to store the full-precision activation map during the inference stage, therefore their memory occupation is significantly larger than the pure 1-bit solution like the vanilla BNN.

## 3 GEOMETRIC ANALYSIS

A typical full-precision neural network block can be described by

$$x^{i+1} = \text{ReLU}(W^i x^i + b^i)$$
$$W^i \in \mathbb{R}^{m \times n}, b^i \in \mathbb{R}^m, x^i \in \mathbb{R}^n, x^{i+1} \in \mathbb{R}^m. \tag{1}$$

Neural networks are trained using the back-propagation algorithm. Back propagation is composed of two components i) forward pass and ii) backward propagation. In the forward pass, the loss function $\mathcal{L}(.)$ is evaluated on the current weights, and in backward propagation, the weights are updated sequentially.

### 3.1 FORWARD

To simplify the analysis, we assume that all weight vectors $W_j^i$ have unit norm. It is a reasonable assumption when the network has batch normalization layers in which all neuron responses are

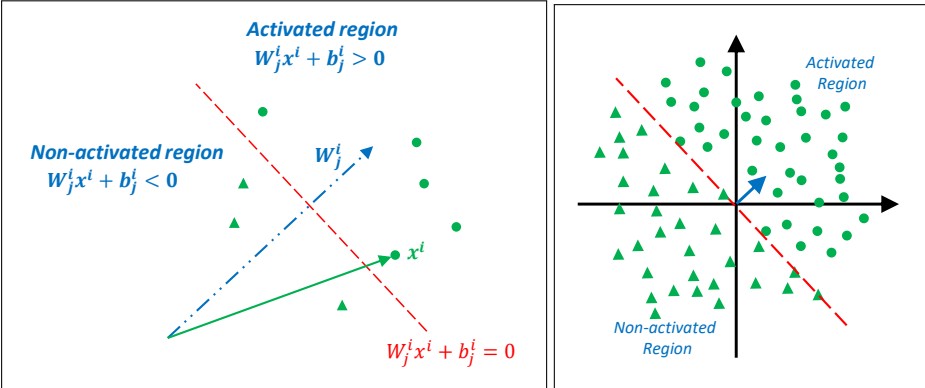

Figure 1: Activated and non-activated regions of ReLU (left panel). Activated region of ReLU at initialization (right panel).

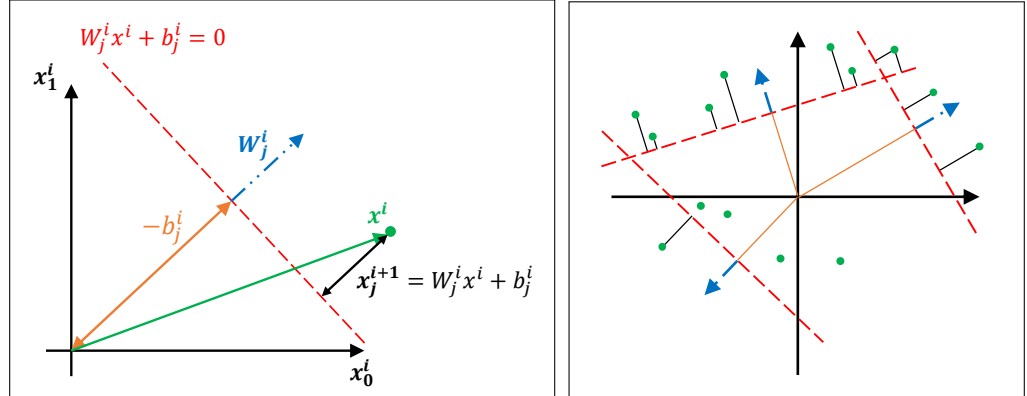

Figure 2: Geometric behavior of ReLu during forward pass, trained hyperplanes (left panel) and their geometry (right panel).

normalized, as the magnitude of the weight vectors does not affect the layer output. The $j^{th}$ neuron response in the $(i+1)^{th}$ layer are computed as

$$x_j^{i+1} = \begin{cases} W_j^i x^i + b_j^i & W_j^i x^i + b_j^i > 0 \\ 0 & W_j^i x^i + b_j^i \leq 0 \end{cases} \tag{2}$$

First, the input data points $x^i$ are projected to the $j^{th}$ row vector of the weight matrix. The dot product of $W_j^i$ and $x^i$ are cut by the corresponding bias term $b_j^i$, i.e. the output $x_j^{i+1}$ is set to zero if the dot product is smaller than the threshold, see Figure 2 (left panel). A hyper-plane whose normal direction defined by $W_j^i$ divides the input space into two parts: i) activated region (non-saturated regime) and ii) non-activated region (saturated regime), see Figure 1. If the data point $x^i$ falls on the positive side of a hyper-plane (activated region), the hyper-plane is activated by $x^i$. Consequently, $x_j^{i+1}$ is positive. Otherwise, $x_j^{i+1}$ equals zero and remain deactivated.

The weight matrix $W^i$ of size $m \times n$ and the bias vector $b^i$ of size $m \times 1$ define $m$ hyper-planes in the $n$-dimensional input space, see Figure 2 (right panel) .

## 3.2 BACKWARD

During backward propagation, the backward gradient update on $W_j^i$ and $x^i$ are computed using

$$
\begin{cases}
\frac{d\mathcal{L}}{dW_j^i} & = \frac{d\mathcal{L}}{dx_j^{i+1}} * \frac{dx_j^{i+1}}{dW_j^i} \\
\frac{d\mathcal{L}}{db_j^i} & = \frac{d\mathcal{L}}{dx_j^{i+1}} * \frac{dx_j^{i+1}}{db_j^i} \\
\frac{d\mathcal{L}}{dx^i} & = \frac{d\mathcal{L}}{dx_j^{i+1}} * \frac{dx_j^{i+1}}{dx^i}
\end{cases}
\tag{3}
$$

For the case of ReLU activation

$$
\frac{dx_j^{i+1}}{dW_j^i} = \begin{cases} x^i & W_j^i x^i + b_j^i > 0 \\ 0 & W_j^i x^i + b_j^i \leq 0 \end{cases}
\tag{4}
$$

$$
\frac{dx_j^{i+1}}{db_j^i} = \begin{cases} 1 & W_j^i x^i + b_j^i > 0 \\ 0 & W_j^i x^i + b_j^i \leq 0 \end{cases}
\tag{5}
$$

$$
\frac{dx_j^{i+1}}{dx^i} = \begin{cases} W_j^i & W_j^i x^i + b_j^i > 0 \\ 0 & W_j^i x^i + b_j^i \leq 0 \end{cases}
\tag{6}
$$

The activation function only allows the gradients from data point on the activated region to backward propagate and update the hyper-plane (equation 4).

## 3.3 RELU ACTIVATION

From the hyper-plane analysis, we realize that ReLU activation has three ideal properties that are distinguishing it from the others i) the diversity of activated regions at initialization, ii) The equality of data points at initialization, iii) The equality of hyper-planes at initialization. These may explain why ReLU activation outperforms the traditional Hyperbolic tangent or sigmoid activations. To argue each property, let us suppose that the distribution of the dot products is zero-centered. This assumption is automatically preserved in the batch normalization. Weight initialization techniques, like Xavier and MSRA, randomly initialize the weights to maintain the output variance.

i) Region diversity: the activated regions of hyper-planes solely depend on the direction of the weight vector, which is randomly initialized. This allows different hyper-planes to learn from a different subset of data points, and ultimately diversifies the backward gradient signal. ii) Data equality: an arbitrary data point $x^i$, is located on the activated regions of approximately half of the total hyper-planes in layer $i$. In other words, the backward gradients from all data points can pass through the approximately same amount of activation function, update hyper-planes, and propagate the gradient. iii) Hyperplane equality: an arbitrary hyper-plane $W_j^i$, is affected by the backward gradients from approximately $50\%$ of the total data points. All hyper-planes on average receive the same amount of backward gradients. Hyper-plane equality speeds up the convergence and facilitates model optimization, see Figure 1 (right panel).

## 3.4 HTANH ACTIVATION

Similar to the ReLU activation, only the backward gradients from data points located in the activated region of a hyper-plane can backward propagate through the activation function and update this hyper-plane. This analysis also applies to htanh activation. The performance gap between ReLU activation and htanh activation is caused by their different activated region distribution, see Figure 3. Clearly, htanh activation is not as good as ReLU in defining balanced and fair activated regions. However, we analyze each property for htanh as well. i) Region diversity: activated regions of htanh are not as diverse as ReLU. Activated regions of htanh cover only the area close to the origin. Assuming Gaussian data, this is a dense area that the majority of data points are located in. ii) Data equality: data points are not treated fairly htanh activation function. Data points that closer to the origin can activate more hyper-planes than the data points far from the origin. If the magnitude

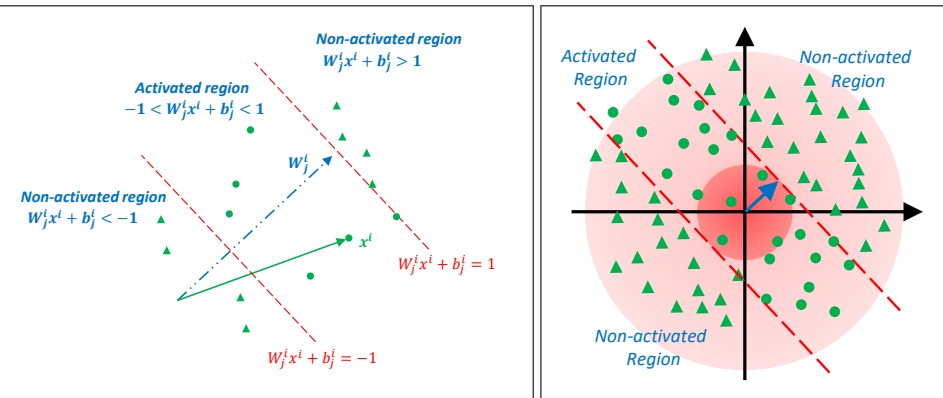

Figure 3: Activated region and non-activated region of htanh activation function(left panel). Activated region of Hard Tanh at initialization (right panel)

of a data point $x^i$ is small enough, it can activate all hyper-planes in the same layer, see the deep-red region of Figure 3 (right panel). As a consequence, in backward gradients, few data instances affect all hyper-planes. In other words, the backward gradients from a part of the training data points have a larger impact on the model than the others. We considered this imbalance ultimately affects model generalization problem since the model training focuses on a subset of the training data points close to the origin. iii) Hyperplane equality: The initial activated regions should cover a similar-sized subset of the training data points overall, and this property is shared in both ReLU and htanh activations.

Similar analysis also applies to other activation functions with the zero-centered activated region, like Sigmoid or Hyperbolic Tangent.

## 4 BIAS INITIALIZATION

Here we proposed a simple initialization strategy to alleviate the data inequality issue and improve activated region diversity for the htanh activation relying on our geometric insight. We argue bias initialization with a uniform distribution between $[-\lambda, \lambda]$, where $\lambda$ is a hyper-parameter is a solution to region diversity. With random bias initialization, the data points that far from the origin can activate more hyper-planes. If $\lambda > \max(\|x\|) + 1$, all data points activate approximately the same number of hyper-planes during backward propagation, so data equality can be achieved. Also, with the diverse initial activated region, different hyper-planes learn from different subset of training data points.

However, this initialization strategy comes with a drawback. Hyper-plane equality no longer holds when the biases are not set to zero. Hyper-planes with larger initial bias have less activated data. Therefore, choosing the optimal value of $\lambda$ is a trade-off between the hyper-plane equality and the data equality. Experiments below shows that the validation curve becomes unsteady if $\lambda$ value set to too high. Empirically, with a batch normalization layer, $\lambda \approx 2$ provide a good initial estimate. In this case, the activated regions covering from $-3$ to $+3$, so it allows the gradients from almost all data points to propagate. It's worth to mention that the experiments showed that small $\lambda$ also helps to improve the performance of ResNet architecture.

## 5 EXPERIMENTS

The proposed bias initialization method is evaluated on the CIFAR-10. The network architectures are based on the original implementation of the BNN (Courbariaux et al., 2016). We choose the VGG-7 architecture and the ResNet architecture.

The VGG-7 architecture, is a simple and over-parameterized model for CIFAR 10. This is an ideal architecture to compare the performance between different activations. In the original implementation, this full-precision architecture is designed to compare with BNN, so the BatchNorm layers are

| Activations | VGG-7 | ResNet |
|---|---|---|
| ReLU (Baseline) | 6.98 | 9.45 |
| htanh | 10.91 | 10.63 |
| htanh ($\lambda$=0.5) | 9.99 | 9.87 |
| htanh ($\lambda$=1.0) | 9.15 | 10.47 |
| htanh ($\lambda$=1.5) | 8.36 | 10.13 |
| htanh ($\lambda$=2.0) | 7.98 | 10.23 |
| htanh ($\lambda$=2.5) | **7.83** | **9.84** |

Table 1: Validation error rate % for full-precision training, $\lambda = 0$ coincides with common deterministic initialization.

| $\lambda$ | Binary VGG-7 | Binary ResNet |
|---|---|---|
| 0.0 | 10.77 | 23.11 |
| 0.5 | 9.57 | 22.31 |
| 1.0 | 9.17 | 22.83 |
| 1.5 | 8.57 | 22.56 |
| 2.0 | 8.56 | 22.47 |
| 2.5 | 8.53 | **22.19** |
| 3.0 | **8.48** | 22.30 |

Table 2: Validation error rate % for Binary training, $\lambda = 0$ coincides with common deterministic initialization.

inserted after each ReLU activation to match the pattern of BNN. This arrangement has a negative impact on the performance of the full-precision neural network. In our full-precision experiments, we put the BatchNorm layers back to their original position, so the error rate of the baseline model is improved from the original reported $9.0\%$ to $6.98\%$. This is close to $6.50\%$, which is the best CIFAR 10 error rate reported on VGG-9 architecture that includes BatchNorm. Considering VGG-9 has much more parameters than this architecture, we keep it as the baseline for full-precision VGG. We also followed the training recipe from the original implementation, SGD optimizer with 0.9 momentum and weight decay set to $5 \times 10^{-4}$. We use more training epochs to compensate for the slow convergence caused by htanh activation.

Figure 4 confirms that the random bias initialization strategy helps to reduce the performance gap between htanh and ReLU activation. A similar effect is observed for ResNet type architectures.

We also tested the proposed bias initialization on the ResNet-like architecture. The results are depicted in Figure 4 re-assures that bias initialization improves htanh and pushes it toward ReLU accuracy, see Table 1.

Binary training that use STE is similar to htanh activation. We expect to observe a similar effect in BNN training with STE gradient approximator. The validation error rate is summarized in Table 2. In the Binary VGG-7 experiments, we reduced the accuracy gap between full-precision network

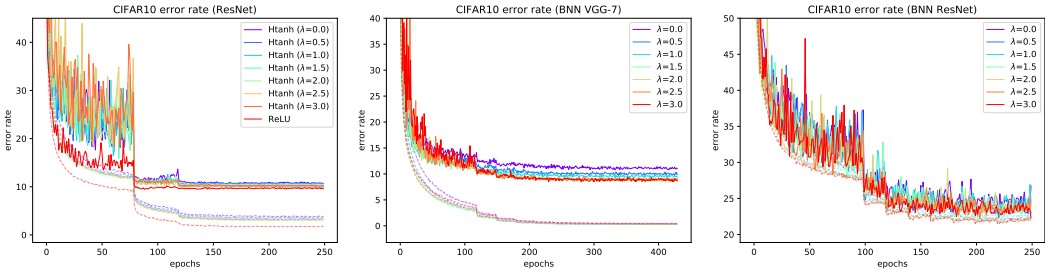

Figure 4: Training of full-precision ResNet architecture (left panel) Binary VGG-7 architecture (middle panel), and Binary ResNet (right panel)

with ReLU activation and BNN from 4% to 1.5%. The bias initialization strategy is effective to close the gap on binary ResNet architecture by almost 1%, even while the full-precision model even under-fits on CIFAR10 data.

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
