# OpenReview forum: "Random Bias Initialization Improving Binary Neural Network Training"
_ICLR.cc/2020/Conference — Reject_

### Official Review · AnonReviewer1 · 2019-10-13
**Official Blind Review #1**

**Rating:** 1

**Review:**

The paper proposes a method for bias initialization and shows that it improves training for BNN.

I vote to reject the paper. Main points against are: (1) is no theory and very limited experiments (2) Bad writing.

Detailed remarks:
 - The level of english is not good enough all over the paper, example: "It is more common to use low-bit quantized networks such as Binary Neural Networks (BNNs)״ more common then what? (I also disagree on the scientific claim)
- The authors claim that XNOR nets and such have "memory occupation is significantly larger than the pure 1-bit solution like the vanilla BNN.". While this is true for training, it is not true for inference which is in many cases where one needs to use limited hardware.
- The paper main claim is the data equality and hyperplane equality are the main strengths of ReLU, but doesn't give any justification or even intuition into why this is the case. I am not convinced that these points are important, and the paper did nothing to try to persuade me.
- Data point equality shouldn't hold for ReLU networks with non-zero bias initialization as well.
- The experiments show promising results but only on cifar10 and only with the outdated BNN, also as a necessary baseline it would be important to show the effect of the bias initialization on ReLU networks.


I believe the paper shows promising initial results but needs to strengthen them considerably. It also needs to improve the writing. A better justification for the method, even if it only at an intuitive level would help considerably.

**Experience Assessment:**

I have published one or two papers in this area.

**Review Assessment: Checking Correctness Of Derivations And Theory:**

N/A

**Review Assessment: Checking Correctness Of Experiments:**

I assessed the sensibility of the experiments.

**Review Assessment: Thoroughness In Paper Reading:**

N/A

---

### Official Review · AnonReviewer2 · 2019-10-20
**Official Blind Review #2**

**Rating:** 1

**Review:**

Summary: This paper tries to improve the training for the binary neural network.

Weaknesses:
[-] A lack of related works. There have been many related works about BNN in these years (after 2017), but the authors do not have a quick summary of them.
[-] More reference. e.g, when authors mention 'many related works require to store the full-precision activation map during the inference stage',  some reference is necessary.
[-] Weak Motivation: The authors argue 'We analyze the behaviour of the full-precision neural network with ReLU activation' in the abstract. However, in Section 3, I cannot find any analysis. Only writing down the backward and forward cannot be called analysis. Initialization is different from the training dynamics. Assumptions and theorems should be highlighted.
[-] Poor writing: A lot of typos. Only in the last paragraph in Section 2, I find many typos,  e.g. 'replaced replacing ReLU activation', 'any relaated works'.

Questions:
[.] In experiments, what structure is used for ResNet? ResNet-18-like or ResNet-110-like? (The results for these two kinds of structure are totally different for binary neural network, as the difference in the number of channels)
[.] In experiments, the performance of the baselines seems lower than related papers? Do the authors increase the number of channels in each layer as the other people do? It can improve the result a lot, and I wonder whether the improvement still exists in this setting.
[.] In experiments, only CIFAR10 results have been reported, but I wonder what is the error bar looks like? (Do the authors run the experiments several times and calculate the variance?)


**Experience Assessment:**

I have read many papers in this area.

**Review Assessment: Checking Correctness Of Derivations And Theory:**

I carefully checked the derivations and theory.

**Review Assessment: Checking Correctness Of Experiments:**

I carefully checked the experiments.

**Review Assessment: Thoroughness In Paper Reading:**

I read the paper at least twice and used my best judgement in assessing the paper.

---

### Official Review · AnonReviewer3 · 2019-10-23
**Official Blind Review #3**

**Rating:** 3

**Review:**

This paper proposes a method to initialize the bias terms in neural network layers, and argues that the proposed method improve the performance of binary neural networks (BNNs). The paper justifies the proposed method by analyzing the geometric properties of the ReLU and the hard tanh (htanh) activation functions, as well as by empirical results on the CIFAR-10 dataset using the (binary variants) of VGG-7 and ResNet.

While closing the performance gap between BNNs and their full-precision counterparts is an interesting problem of practical importance, this paper has several limitations:

(1) the analysis of geometric properties of ReLU/htanh is not sufficiently precise and clear;
(2) the paper does not clearly present the connections between the htanh activation function and the straight-through estimator employed in back-propagating the gradients in training a BNN;
(3) the experimental results are too limited on just one dateset, and only error rate on validation set is reported, however, lower error rate on validation set won't guarantee better performance on test set;
(4) the presentation is imprecise and unpolished.


Minor comments:

Section 2:
"Tang et al. replaced replacing ReLU" -> "Tang et al. replaced ReLU"
"many relaated works" -> "many related works"

Section 3:
please define the symbols used in Equation (1)
title of Figure 2: "behavior of ReLu" -> "behavior of ReLU"

**Experience Assessment:**

I do not know much about this area.

**Review Assessment: Checking Correctness Of Derivations And Theory:**

I carefully checked the derivations and theory.

**Review Assessment: Checking Correctness Of Experiments:**

I assessed the sensibility of the experiments.

**Review Assessment: Thoroughness In Paper Reading:**

I read the paper at least twice and used my best judgement in assessing the paper.

---

### Decision · Program_Chairs · 2019-12-19

**Decision:**

Reject

**Comment:**

The article studies the behaviour of binary and full precision ReLU networks towards explaining differences in performance and suggests a random bias initialisation strategy. The reviewers agree that, while closing the gap between binary networks and full precision networks is an interesting problem, the article cannot be accepted in its current form. They point out that more extensive theoretical analysis and experiments would be important, as well as improving the writing. The authors did not provide a rebuttal nor a revision.